# Encapsulation: A Strategy to Deliver Therapeutics and Bioactive Compounds?

**DOI:** 10.3390/ph16030362

**Published:** 2023-02-27

**Authors:** Iveta Klojdová, Tomáš Milota, Jitka Smetanová, Constantinos Stathopoulos

**Affiliations:** 1DRIFT-FOOD, Faculty of Agrobiology, Food and Natural Resources, Czech University of Life Sciences Prague, 165 21 Prague, Czech Republic; 2Department of Immunology, Second Faculty of Medicine Charles University and Motol University Hospital, 150 06 Prague, Czech Republic

**Keywords:** encapsulation, targeted therapy, targeted distribution of therapeutics, drugs, vaccines, food supplements, functional foods, immunogenicity

## Abstract

There is a growing demand for efficient medical therapies without undesired side effects that limit their application. Targeted therapies such as deliveries of pharmacologically active compounds to a specific site of action in the human body are still a big challenge. Encapsulation is an effective tool for targeted deliveries of drugs and sensitive compounds. It has been exploited as a technique that can manage the required distribution, action and metabolism of encapsulated agents. Food supplements or functional foods containing encapsulated probiotics, vitamins, minerals or extracts are often part of therapies and currently also a consumption trend. For effective encapsulation, optimal manufacturing has to be ensured. Thus, there is a trend to develop new (or modify existing) encapsulation methods. The most-used encapsulation approaches are based on barriers made from (bio)polymers, liposomes, multiple emulsions, etc. In this paper, recent advances in the use of encapsulation in the fields of medicine, food supplements and functional foods are highlighted, with emphasis on its benefits within targeted and supportive treatments. We have focused on a comprehensive overview of encapsulation options in the field of medicine and functional preparations that complement them with their positive effects on human health.

## 1. Introduction

Considering the increasing age of the population and other factors such as lifestyle (workload, stress, environmental pollution, etc.), there is a global effort to ensure a certain standard of living, with an emphasis on the increasing level of medical care as well as on a healthy and nutritionally balanced diet [1,2,3]. Many people in developed countries suffer from obesity, a complex multifactorial disease. For adults, guidelines from the US Centers for Disease Control and Prevention and the WHO (World Health Organization) define a normal body mass index (BMI) range as 18.5 to 24.9, whereas a BMI ≥ 25 is considered to be overweight and a BMI ≥ 30 is classified as obese. Severe obesity is defined as a BMI ≥ 35. High BMI increases the risk of type 2 diabetes, hypertension, cardiovascular disease and some cancers [4,5]. In all of these cases, an effective delivery of therapeutic agents and nutrients plays the most important role. The appropriate method of administering therapeutic agents ensures their required effect on human health and minimal side effects [6,7]. Food supplements should also be administered in a way that guarantees their supportive effects on the human body. Moreover, many consumers are concerned with the nutritional aspects of food and interested in novel functional food products containing plant extracts, vitamins, probiotics and prebiotics, etc. Almost all of these active components are very sensitive compounds, and without protective treatments they lose their functional properties [8]. An effective tool for delivery of therapeutic agents and sensitive compounds is encapsulation, which ensures not only protection and defined distribution in some organs of the human body but also may change the metabolization of some agents [9,10]. The changes in metabolization of the agents may be controlled by the presence of other components in the protective layer [11]. Encapsulation can be defined as the effective surrounding of an agent that provides its protection. The encapsulated sensitive agents (pure substances or a mixture of more agents) are known as coated materials, core materials, payloads, internal phases, etc., while the coating materials can be called shells, wall materials, capsules, carriers, membranes, films, the outer shell or packing materials [12]. This technique enables managing the metabolic processes of encapsulated agents and their delivery to the specific site of action. Thus, encapsulation can ensure the effective distribution of drugs and nutrients in the human body. In general, the administration of poorly soluble, toxic or sensitive drugs is improved using encapsulation. Additionally, side effects can be limited [6]. The most common route of encapsulated agent administration is oral administration [13,14]. However, other routes of administration (intravenous, subcutaneous, dermal etc.)—e.g., implantations of encapsulated cells into tumours—have been published [15]. The summary of encapsulation use is shown in Figure 1. Generally, the creation of encapsulated cells is a form of cell surface modification (e.g., an entrapment within a biopolymer structure) and has drawn a large amount of attention in various research areas, such as cell therapy, cell biosensors, biocatalysts, etc. [16]. Because of growing demand for end products made using encapsulation, new techniques and methods have been developed. The choice of preparation method for an agent’s encapsulation depends mostly on its properties, such as state of aggregation, sensitivity, size of molecules, etc., as well as the method of administration [17]. Here, we review recently published new encapsulation approaches with high potential for use in the field of advanced therapies in medicine and targeted functional foods. This review was prepared in line with the preferred reporting items for systematic reviews and meta-analyses guidelines [18] and the proposed guidelines for biomedical narrative review [19].

## 2. Encapsulated Agents

### 2.1. Medicine

In medicine, many encapsulation methods have been developed by using new materials and various types of drug–carrier interactions [6]. The use of encapsulation methods in medicine is very traditional and has been studied intensively. Here, new approaches are based not only on drug encapsulation but also on advanced cell encapsulation [15].

#### 2.1.1. Drugs

Encapsulation of drugs is a vital strategy for poorly soluble, fragile or aggressive compounds and can achieve a stronger therapeutic effect along with minimized side effects [6]. However, producing formulations with high encapsulation efficiencies of (bio)active compounds remains a big challenge [20]. Encapsulation into, e.g., liposomes facilitates drug delivery to a specific site (such as brain parenchyma), enhances the stability of drug molecules, prolongs pharmacological activity via continuous local release of active molecules and reduces side effects, improving the effectiveness and safety of therapies [15]. Due to the huge number of drugs with very specific and different properties, many types of biopolymers (protein-based particles) and artificial polymers (polylactic-co-glycolic acid, poly-ε-caprolactone, polylactic acid, poly(butyl cyanoacrylate or gelatine) are used for their encapsulation [21,22]. Additionally, many encapsulated active molecules have been commercialized in the pharmaceutical market, including anti-cancer (e.g., doxorubicin, paclitaxel, vincristine), hormonal (leuprolide), antiparasitic/antifungal (amphotericin) or analgesic therapeutics (morphine) [21]. Not only biopolymers or artificial polymers can be used for encapsulation of active molecules in the field of pharmacy. New approaches have been reported—e.g., the use of erythrocytes as ”capsules” for drugs [23]. In the last decade, encapsulation of drugs for anti-cancer therapies [24] as well as nanoparticles [25,26] have attracted significant interest.

#### 2.1.2. Cells

Despite consistent increase in the mean life expectancy of the population, healthy life expectancy has not increased. The replacement and reconstitution of diseased or damaged tissues has become an immense challenge [27]. Here, cell encapsulation has become a promising strategy. Essentially, live cells are entrapped within a semipermeable membrane. There are two basic approaches: encapsulation of a cell mass or of a single cell. The encapsulation of a cell mass is the covering of a cell group with the appropriate material in order to evaluate the cellular function as a whole. On the other hand, single-cell encapsulation entails surface-coating individual cells with some material. Single-cell encapsulation has become very attractive in recent years and has a great amount of potential in treatments for serious diseases such as cancers [16]. Encapsulation, where biopolymer gels are usually used, represents an evolving branch of biotechnology and regenerative medicine [28]. Different types of encapsulated cells with protein secretions have been investigated for disease treatments: islet cells (insulin secretion)—diabetes treatment [29], kidney cells (hepatic lipase secretion)—hyperlipidaemia treatment [30], myoblasts [31] and fibroblasts [32] (human factor IX secretion)—haemophilia treatment, etc. Cell encapsulation is also part of regenerative medicine [27]. Encapsulation of mammalian cells has been used in the regeneration of different tissues: skeletal muscle as engineered tissue—encapsulated myofibroblasts [33], dermis as engineered tissue—encapsulated fibroblasts [34], bone as engineered tissue—encapsulated osteosarcoma cells [35], etc. In recent years, the potential use of encapsulated cells for clinical application in malignant brain tumour treatment has also been discussed [15].

### 2.2. Food Supplements and Functional Foods

Nowadays, functional foods and supplements are very popular products. There are a lot of sensitive compounds that can be encapsulated using different techniques. Products so prepared support human health and are an integral part of the modern lifestyle [36]. Here, we present the most-frequently encapsulated sensitive materials.

#### 2.2.1. Vitamins

Generally, vitamins possess important physiological functions, including antioxidative, immunoregulatory, anti-inflammatory, etc. Unfortunately, these chemical structures are highly sensitive to high temperatures, light, oxygen and extreme pH conditions [37]. Encapsulation enables preserving their bioactive properties and can effectively improve the administration of sensitive vitamins and their metabolization. Hydrophobic (fat-soluble) vitamins, namely A, D, E and K, are provided mainly by foods. They can be added in functional foods, helping to treat skin diseases and several types of cancer or decrease oxidative stress [38]. The encapsulation efficiency of polymers was in the range of 27–45% [39]. Additionally, the prepared capsules with hydrophobic vitamins ensure thermal stability of these sensitive molecules up to 170 °C, which enables further manufacturing of functional products [40]. Enhanced stabilities and gradual releases of these hydrophilic vitamins have also been reported. Vitamin B has been encapsulated as a component of plant extracts [41] and the improved gradual release of vitamin C has been observed when encapsulated within multiple emulsion [42] or casein gels [43].

#### 2.2.2. Probiotics

In recent years, encapsulation of probiotics and their use in functional supplements and foods have been studied intensively. According to the definition provided by the WHO, probiotics are live organisms, such as bacteria and yeasts, which furnish health benefits to the host [44]. The systems with encapsulated probiotics are very specific and the viability of microorganisms is influenced by many factors: the encapsulation method itself, the materials used for encapsulation, exposure to oxygen, low pH, digestive enzymes, heat treatment and microbial strains [45,46]. The viability of probiotics may be greatly improved by co-encapsulation with prebiotics (materials which support the growth of probiotics) [47]. Products containing probiotics and prebiotics, representing a synergistic effect, are called synbiotics [48]. The most common probiotics are Gram-positive genera such as strains of *Lactobacillus*, *Bifidobacterium*, *Leuconostoc*, *Pediococcus*, *Enterococcus*, *Streptococcus* and *Bacillus* [49]. Products with encapsulated probiotics may contain either a single strain or a mixture of two or more strains. The health effects of probiotics are strain-specific. A single strain may offer different benefits when used individually (e.g., specific action against a certain bacterium) and in combination with other strains (general support of digestion) [50]. Consumption of functional products with probiotic contents have been associated with stimulation of the immune system (better resistance to infections of the respiratory and gastrointestinal tracts [51] and prevention of colorectal cancer and improvement of inflammatory bowel disease, among others [52]. These benefits of a probiotic formulation also differ with the patient group (age, general condition of the patient, etc.) [50]. However, conflicting clinical studies on the undesired effects of probiotics have also been published [53]. The use of probiotics has been associated with a higher risk of infection and/or morbidity in young infants and postoperative, hospitalized or immuno-compromised patients, in part due to bacteraemia and fungaemia [54]. Nevertheless, most probiotics are recognized as food-grade and are recommended for human consumption by regulatory authorities; encapsulation is an effective tool and a great way to improve their efficient delivery in target sections of the gastrointestinal tract [52,53].

#### 2.2.3. Extracts

Many plants are a natural source of various compounds with diverse biological activities that can improve treatment of some diseases. Extracts are complex mixtures with high contents of antioxidant, antibiotic, antiviral, anticancer, antiparasitic, antifungal, hypoglycaemic, anti-hypertensive and insecticidal properties. The extraction of the extracts is usually provided by organic solvents and the resulting extracts are sensitive and unstable [55]. Encapsulation is a way to overcome this problem. The enhanced stabilities of encapsulated active substances from plant extracts have been described by many authors: elderberry extract [56], agro-industrial by-product extracts [57], Mediterranean plant extracts [58], etc. Some plant extracts rich in polyphenols, alkaloids and terpenoids have been found as efficient materials for the preparation of food supplements with beneficial anti-obesity effects [59].

## 3. Encapsulation Methods and Techniques

Many methods can be used for the preparation of encapsulated agents, according to the application intent. The spray-drying and freeze-drying techniques are the most frequently used techniques for the encapsulation of sensitive drugs (pharmaceutics), sensitive food supplements, etc. The main advantages are the low thermal stress and a liquid feed becoming powder in one step. For drug delivery, the solubility of powders with encapsulated sensitive compounds prepared using spray-drying is too low. This is the main reason why the technique is mainly used for manufacturing inhaled drugs [60,61]. Recently, a new unique technique of coating capsules with a powder layer has been used. The additional powder coating enables the advanced adjustment of the controlled release of the encapsulated agent [62].

For many of these preparations (drugs, food supplements), spray-drying and (spray) freeze-drying techniques are the final steps in the manufacturing process. However, the manufacturing of functional foods can involve other steps, such as mixing with other components (fats, liquid matrices, etc.), heat treatment, etc. The main areas of functional food application are dairy products, juices, bakery products, etc. [63].

An overview of the most common systems and methods used for encapsulation is provided in Figure 2. Recently published preparations of some promising systems with encapsulated agents using different techniques are given in Table 1.

### 3.1. Capsules

Capsules are probably the most common system for encapsulation. Their composition must ensure the protection of the encapsulated sensitive compound as well as the ability to release the encapsulated compound in the appropriate sites in the intestine [78]. Encapsulation in capsules—within single or multiple biopolymer coatings—was published as an effective tool for the protection of potential probiotic strains against the undesired effects of stomach acid and bile acid during digestion [79], as well as for the protection of cells [80]. The most commonly used materials for capsules include hydrocolloids, proteins, starches, dextrins, lipids, various emulsifiers and fibres, alone or associated with other compounds. The choice of the most appropriate encapsulation technique depends on the material to be encapsulated and the purpose of use, and is limited by the availability of equipment and intended capsule sizes [81]. Capsules with sensitive bioactive compounds are often prepared using spray-drying techniques. In this type of technique, a bioactive compound mixture is atomized into a wall material under a hot-air current. After instant drying and spraying of the mixture, the resulting product, in powder form, consists of the bioactive compound covered by wall material [82]. A modification with a cool air flow is also possible. This type of technique is then called spray cooling or chilling and is often used when the barrier material consists of lipids [83]. Some examples of capsule encapsulation published in the past five years are shown in Table 2. Cell encapsulation remains a big challenge for bioengineers. The main concept of these capsules is a preparation based on hydrocolloids (usually alginate) with specific surface properties to interact with target tissues in the human body. However, as intensive as the studies are, there are still no licensed therapies [80].

### 3.2. Emulsions

Emulsions are widely used in the fields of medicine, pharmacology and food systems. Many emulsion-based delivery systems for polyphenols have been well established, including single and multiple emulsions. Both types of emulsions (according to their complexity) can be additionally considered as macro-, micro- (less than 100 or 200 nm) and nano-emulsions (typically 20–200 nm) [92,93,94]. Emulsions can be stabilized by emulsifiers or Pickering particles (Pickering emulsions) or by a combination of both these stabilization agents (co-stabilized emulsions) [95,96]. Emulsions can also be used as the basic mixture for preparations of different types of encapsulation systems—e.g., capsules manufactured using spray-drying methods [40].

#### 3.2.1. Simple Emulsions

Commonly used simple emulsions are water-in-oil (w/o) and oil-in-water (o/w). Hydrophobic emulsifiers are used for preparations of w/o emulsions and hydrophilic emulsifiers for preparations of o/w emulsions. They are mostly further manufactured during preparations of advanced systems for encapsulation [97].

#### 3.2.2. Multiple Emulsions

Encapsulation of drugs and other bioactive compounds in multiple emulsions provides more complex solutions than encapsulation in simple emulsions. Multiple emulsions (MEs) are complex structures composed of multiple water and oil phases. Because of their high complexity, these systems are thermodynamically unstable systems and have a strong tendency for phase separation. The osmotic pressure of internal and external phases must be balanced to avoid any instabilities such as internal droplet shrinkage or growth [98,99]. The stabilization of multiple emulsion systems with proper agents, such as emulsifiers, Pickering particles and other compounds, is also crucial to avoid coalescence [100]. Generally, double systems of emulsions have been prepared: water-in-oil-in-water (w/o/w), oil-in-water-in-oil (o/w/o) and less often solid-in-oil-in-water (s/o/w) [98]. W/o/w multiple emulsions are usually used for the encapsulation of sensitive hydrophilic compounds and o/w/o multiple emulsions for the encapsulation of sensitive hydrophobic compounds [101]. Multiple emulsions can be prepared by various methods. The most frequent methods are two-step emulsifications, where simple (internal) emulsions are created first and are consequently emulsified into an external water or oil phase (based on emulsion type). Many techniques such as rotor–stator homogenizers, high-pressure homogenizers, membrane units, colloid mills, sonicators, microfluidic devices, etc. are used for emulsification processes [101,102]. Some applications of multiple emulsion as encapsulation system can be found in Table 3. Moreover, w/o/w multiple emulsions are used not only as encapsulation tools, but also as masking agents of undesired taste and as systems which enable the decrease of fat content in foods. Because of these advantages, they have great potential in the field of functional food preparations [99].

### 3.3. Particles

Encapsulation of bioactive compounds (mostly drugs) in particles has advanced significantly in recent years. Microparticles (1 to 1000 µm) can be formulated for controlled administration by almost all routes, with some limitations for intravenous injection. Nanoparticles (1 to 100 nm) can be administered with no limits. Nanoparticles can be much smaller than human cells.

Particles, both micro- and nano-sized, are a promising system for delivery of highly effective drugs to target cells. These particles could be then used, e.g., for modern cancer therapies [112]. Sometimes, particles are considered capsules and vice versa. Particles with required sizes between 1 and 5 μm for inhalation delivery, between 0.1 and 0.3 μm for intravenous delivery and between 0.1 and 100 μm for oral delivery can be prepared by the supercritical antisolvent method [113]. A more common method for manufacturing particles with encapsulated drugs is spray drying or freeze drying (drying of frozen material by sublimation of water under vacuum). It can be preceded by the formation of a matrix, which is consequently sprayed after drying, forming microparticles [81,83]. All these methods can be used for the preparation of either micro- or nano-particles based on the materials used. Some indicative uses are specified in Table 4.

#### 3.3.1. Janus Particles

In recent years, researchers have focused on the potential use of Janus particles for encapsulation. Janus particles have surfaces that have two or more distinct physical properties. The unique surface of Janus particles offers two different types of chemical composition occurring on the same particle. They can contain a magnetic part that permits driving them by means of a magnetic field toward different places in the human body where they can interact [114]. These particles present versatility for a myriad of future applications [115] because of their asymmetry and characteristics suitable for drug delivery, controlled release, diagnostics and even in self-assembly systems [116]. Janus materials can be produced in various anisotropic shapes and chemical compositions [117] using several synthetizing techniques: selective surface modification, seeded crystallization, microfluidics, self-assembly of block copolymers and electrochemical deposition [118].
pharmaceuticals-16-00362-t004_Table 4Table 4Some indicative uses of particles for encapsulation.Encapsulated AgentMicro/NanoIntended UseReferenceResveratrolMicro/Nano (less than 1 µm)Prevention of ageing, cancer, inflammation, neurodegenerative, and cardiac diseases[119]Green tea extractMicroPrevention of ageing[120]QuercetinMicro/Nano (less than 1 µm)Therapeutic agent and a food component[121]Alpha-tocopherol (active form of vitamin E)Micro/Nano (less than 1 µm)Foods, improvement of light, heat and oxygen stability[122]


#### 3.3.2. Liposomes

Liposomes are special kinds of particles. More than half a century after the discovery of liposomes, targeted liposomes or formulations able to deliver a drug after a stimulus. Liposomes have been approved by regulatory agencies [123], however, only a few hydrophilic small-molecule drugs loaded in liposomes with high encapsulation efficiency are available on the market [124]. In general, liposomes are promising systems because they have low toxicity and can encapsulate both hydrophobic and hydrophilic types of compounds [125]. They are therefore supposed to be an effective tool for specific deliveries of antibiotics [123] or essential oils [126]. Liposomal structure is composed of one or more phospholipid bilayers in an aqueous environment [127] and can be produced in the micro and nano ranges [128]. Phospholipids, the major components in liposome structures, contain both hydrophilic (polar head) and hydrophobic (fatty acid chain) groups; however, the mechanism of liposome formation is not yet well known in detail [129]. Conventionally, liposomes are produced using the Bangham method or thin-film hydration, solvent injection, reverse-phase evaporation or detergent removal. New approaches include spray drying, heating, supercritical reverse-phase evaporation, freeze drying and modified ethanol injection [127]. Table 5 summarizes recent published encapsulation using liposomes.

Lipid nanoparticles (LNPs), also called solid lipid nanoparticles, commonly consist of the neutral phospholipids (such as 1,2-distearoyl-sn-glycero-3-phosphocholine, l-α-phosphatidylcholine or 1,2-dioleoyl-sn-glycero-3-phosphocholine) that are essential components of lipid bilayer and cholesterol-enhancing membrane stability. Critical components of LNPs include the ionizable cationic and PEG lipids; both allow high encapsulation efficacy and the steric barrier effect. The charge of ionizable cationic lipids is pH-dependent: they have a positive charge at acidic pH and acquire a neutral charge at physiological pH. Lipid formulations can be prepared by various methods including thin film hydration, detergent depletion, solvent injection, reverse-phase evaporation and emulsion [134]. The size of nanoparticles ranges from 10 to 1000 nm. Incorporated components are located between fatty acid chains, lipid layers or crystal imperfections [135]. LNPs can be engulfed by various mechanisms such as macropinocytosis and clathrin- and caveolae-mediated endocytosis. The form of endocytosis depends on the properties of the nanoparticle and the cell type. LNPs are usually captured in an endosomal compartment system. Endosomal release is a crucial step for effective drug delivery. Essentially, positively charged lipids facilitate fusion with negatively charged endosomal membranes [136]. LNPs thus have the potential to deliver hydrophilic drugs such as chemotherapeutic, antiparasitic or antifungal drugs into the cells or brain through the haematoencephalic barrier. Targeted delivery reduces toxicity and increases drug efficacy [137].

## 4. Behaviour and Distributions of Dispersions in the Human Body

Generally, all encapsulated compounds are supposed to be safely transported to the site of action (for both oral and non-oral methods of administration). Therefore, encapsulation may protect sensitive compounds (e.g., different types of drugs, various substances for functional foods and food supplements, such as polyphenols or probiotics) as well as the surrounding organs, tissues and cells against aggressive substances (e.g., cancer therapies) [24,45,93].

### 4.1. Gastrointestinal Tract

The most common method for administrating encapsulated compounds is oral administration [138]. Essentially, products with encapsulated agents are drunk (drugs are usually swallowed with water or any drink, solid foods are divided into digestible portions in the mouth and swallowed) and are transported to the stomach [139]. Here, the critical point is to ensure the integrity of systems loaded with sensitive compounds in the stomach [10]. The stomach is an integral organ of the gastrointestinal tract that provides size reduction and enzymatic hydrolysis of solid food as well as gastric emptying [140]. The proximal stomach is mostly responsible for the emptying of liquids and works as a reservoir for solid and liquid foods. The distal stomach (antrum) is then the propeller, grinder and siever of solid food [141]. The disintegration rate of foods in the human stomach is strongly influenced by the composition and physicochemical properties of foods as well as by physical forces and chemical reactants (acidic) present in the stomach. This is the crucial step for formulations with encapsulated sensitive compounds that must remain intact [142]. Subsequently, the foods or drugs enter the duodenum. Pancreatic juices containing powerful digestive enzymes are delivered and digestion is finally completed in the small intestine, where many sensitive encapsulated compounds and drugs are released. Here, absorption of nutrients from food takes place. The large bowel ensures dehydration of the gastrointestinal contents [139]. Over the past decades, there have been many in vitro models reported for the whole gastrointestinal tract simulation—models from static bioreactors to multi-compartmental and dynamic systems that can track the structural and physicochemical changes during digestion within the human gastrointestinal tract [143]. In vitro digestion models offer a more technically, ethically and financially available way to study the digestion process than in vivo models [144]. In vivo experiments on humans are usually provided as an end-point measurement with products that are directly applicable to human food consumption and approved by the relevant authorities [140]. In general, almost all studies published previously have been focused on an effective (or managed) release of bioactive encapsulated compounds in the intestine [45,52,88]. Thus, the basic prerequisite for the appropriate function of products with loaded bioactive agents is their stability in an acidic environment and against the proteolytic enzyme protease. Then, they are supposed to ensure the leakage of the encapsulated agent in the intestine, where it is attacked by lipases and pancreatic juices [145]. This process is shown schematically in Figure 3.

### 4.2. Other Methods of Administration

Other methods of non-oral administration of encapsulated compounds may be adapted according to the required treatment and target organs or tissues [146]. Here, we have focused on the most commonly used non-oral means of administration for systems with encapsulated agents or cells (Figure 4). Generally, biomaterial strategies such as the administration of encapsulated sensitive compounds (biological factors, anti-inflammatory drugs, stem cells, etc.) through nanoparticles and capsules (hydrogels) can more effectively treat damaged tissues when injected directly to the site of action [147]. This fact represents a clinical advantage, since therapy with encapsulated agents or cells can reach even difficult targets such as the brain or the eye [148]. For regenerative medicine, the main goal is to deliver encapsulated cells to the damaged area so that they promote the progress in tissue regeneration [149]. Encapsulation of cells also exerts multiple benefits in bone regeneration therapy: the retention of cells on the target site, protecting the cells from mixing and injection forces and conformal filling of the defect shape [150]. For the correct functionality of the system, the selection of the biomaterial used for encapsulation is crucial [151]. One of the most commonly used biomaterials is alginate. However, the average molecular weights of commercially available alginates are high and therefore their modification is often mandatory before their use for regenerative purposes [152]. Another option for administration of encapsulated agents is the subcutaneous route, a relatively easy and clinically applicable administration procedure, which offers slower absorption compared to other parenteral routes and has been successfully studied for the treatment of chronic anaemia with implantation of encapsulated cells that have been able to release erythropoietin [153] or encapsulated cells that have delivered therapeutic antibodies [154]. Intravitreal administration of encapsulated cells for ocular disease treatments have been studied to enable the local treatment of multiple retinal diseases, including age-related macular degeneration or diabetic macular oedema [155]. Encapsulation has also been intensively investigated as a tool for systematic therapies in the intracranial area. A major obstacle for drug delivery in the diseased brain is the passage of molecules between the blood and the brain parenchyma that is regulated by the blood–brain barrier. Implantation of encapsulated therapeutically active cells directly into the brain offers local long-term delivery of therapeutics de novo with reduced side effects [15]. For the implantation of cell microcapsules, the peritoneal cavity also represents an optimal site because of relative safety and good accessibility [156].

### 4.3. Immunogenic Properties

Various delivery systems and materials that have been used for encapsulation also possess immunogenic properties. Particles such as alum-stabilized Pickerings or squalene-based emulsions, liposomes or lipid nanoparticles can be recognized by the immune system and trigger an immune response. These features allow their use as the adjuvants in vaccines [157,158,159]. Alu-stabilized Pickering emulsions (ASPE) consist of alum microgel and a squalene/water interphase. In ASPE, alum tends to adsorb on the interphase, reducing surface tension and improving the stability of the emulsion. Additionally, ASPE showed a higher affinity for dendritic cells (DCs), resulting in a higher humoral and T cell immune response compared to the conventional adjuvants (80). In vaccines, the target antigen (Ag) adsorbs to alum. Alum prolongs retention of Ag at the site of injection and mediates its slow release. Alum also facilitates Ag uptake by DCs that further promote immune response toward antibody-mediated protection directed by T helper (Th) cells [160]. These features assume the use of ASPE as a potent vaccine adjuvant.

MF59 and AS03 represent adjuvant systems based on oil-in-water squalene-based emulsions. MF59 was approved as a component of human influenza vaccine in 1997. The MF59 emulsion is stabilized by Tween 80 and Span 85 [161]. AS03 is an adjuvant system containing squalene, polysorbate 80 and alpha-tocopherol that further enhances immunogenic properties. Similar to MF59, AS03 has also been predominantly used in human influenza vaccines [162]. MF59/AS03 adjuvant systems containing antigens are engulfed by neutrophils and monocytes, which further differentiate into monocyte-derived DCs (Mo-DCs) [163,164]. Mo-DCs initiate an antigen-specific T follicular helper cell response that is essential for B cell isotype switching and production of IgA/IgG-specific antibodies [165].

Liposomes as self-assembling phospholipid bilayer-enclosed spherical particles have shown to have immunostimulatory properties. As a potent adjuvant system, liposomes may deliver multiple Ags or may be combined with other adjuvants or functional molecules to enhance vaccine reactogenicity [166]. For instance, AS01B or QS21 adjuvant systems consist of the triterpenoid saponin and monophosphoryl-lipid A (MPLA). Both adjuvants have been approved for use in malaria (Mosquirix, GlaxoSmithKline Biologicals, Rixensart, Belgium) and shingles vaccine (Shingrix, GlaxoSmithKline Biologicals, Belgium) [167]. MPLA activates DCs via the Toll-like receptor 4-dependent pathway and facilitates generation of TFH, consequently enhancing B cell maturation and production of specific antibodies [168].

The mechanisms of action of saponins as an immunogenic substance have not yet been entirely revealed. Saponins seem to be potent stimulators of cytotoxic T cells and elicit both Th1 and Th2 responses [169,170]. Although the immunogenicity of lipid nanoparticle (LNP)-based mRNA vaccines are boosted by PRRs such as Toll-like receptors (TLR7 and TLR8), RIG-I and cytosolic sensors recognizing mRNA [171,172,173], and LNPs mainly protect and enhance mRNA delivery into the host cell cytoplasm, several studies suggest potential adjuvant activity of LNPs [174,175,176]. The mechanism of action of LNPs is not well described. Recent findings show LNP formulation induces IL-6 secretion and elicits robust Tfh response along with durable humoral response [177]. So far, mRNA vaccines have been tested for influenza and rabies and as anti-cancer vaccines [178]. The largest expansion of mRNA vaccines and LNP-based adjuvant platforms has occurred during the COVID-19 pandemic. A number of the other aforementioned delivery systems are being tested or have been approved for clinical use [179] and may serve as safe and effective adjuvants in anti-SARS-CoV-2 vaccines.

## 5. Conclusions and Future Prospects

Studies on encapsulation and delivery of (bio)active agents or cells reflect the increasing demand for therapeutics with higher efficiency and specificity without undesired effects as well as possible targeted approaches. Encapsulation methods are still being developed and contribute to the new progression of clinical procedures in key areas such as cancer treatment, regenerative medicine, etc. Additionally, many novel approaches in encapsulation technologies for functional food preparations have been reported. Capsules, particles, emulsions, multiple emulsions, liposomes or lipid nanoparticles are further being studied as innovative systems for the administration of sensitive agents such as drugs, probiotics, antioxidants, cells, etc. They have also shown to be potential delivery systems for different vaccines; their immunogenic properties also allow their use as potent and safe adjuvant platforms. Prepared products can thus contribute to the targeted treatment of diseases and prevention of many diseases and be part of a healthy lifestyle. In the future, we expect further development of systems and products with encapsulated agents or cells applied toward tailored therapies and diets. There are still challenges to the wider clinical application of systems with encapsulated agents in personalised therapies.

## Figures and Tables

**Figure 1 pharmaceuticals-16-00362-f001:**
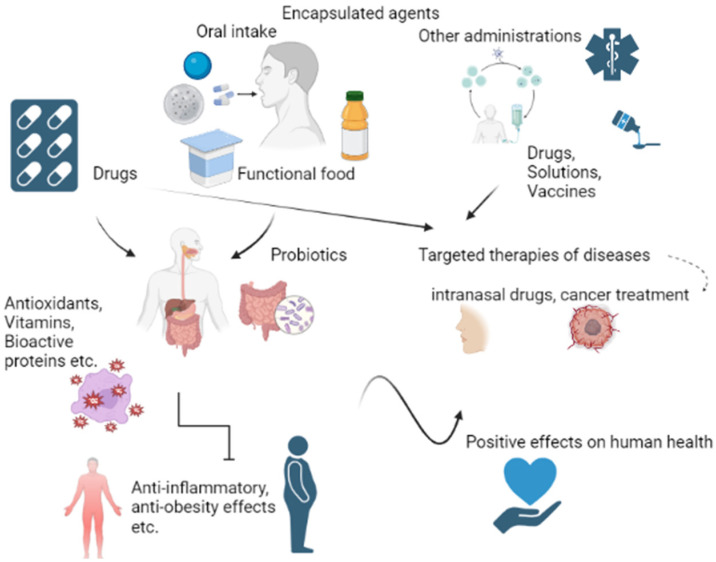
Overview of encapsulation uses.

**Figure 2 pharmaceuticals-16-00362-f002:**
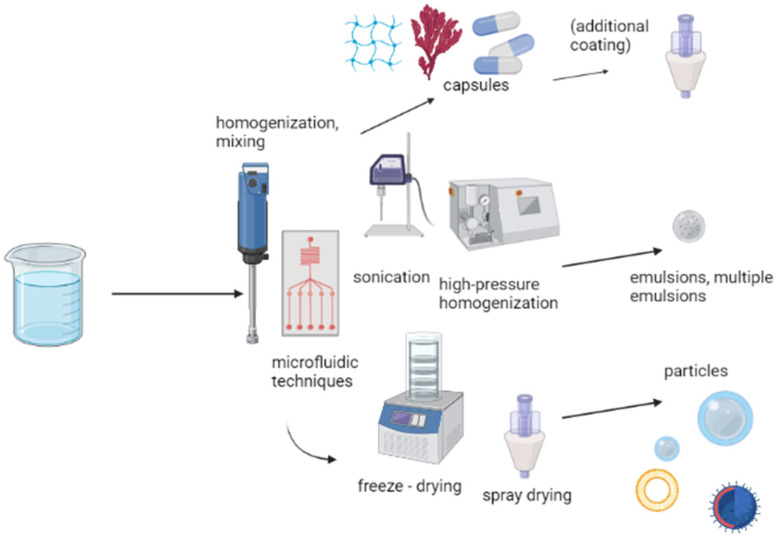
The most common approaches in encapsulation of agents.

**Figure 3 pharmaceuticals-16-00362-f003:**
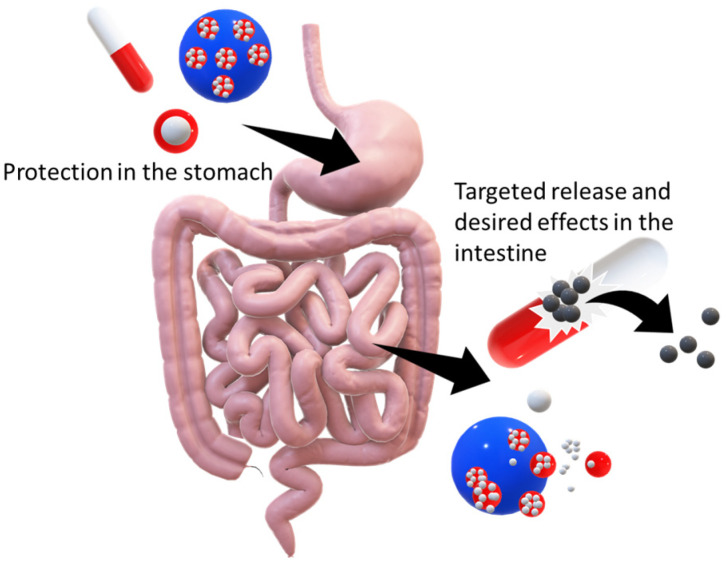
Schematic representation of the behaviour of several systems (capsules, particles and multiple emulsions) with encapsulated compounds in the gastrointestinal tract.

**Figure 4 pharmaceuticals-16-00362-f004:**
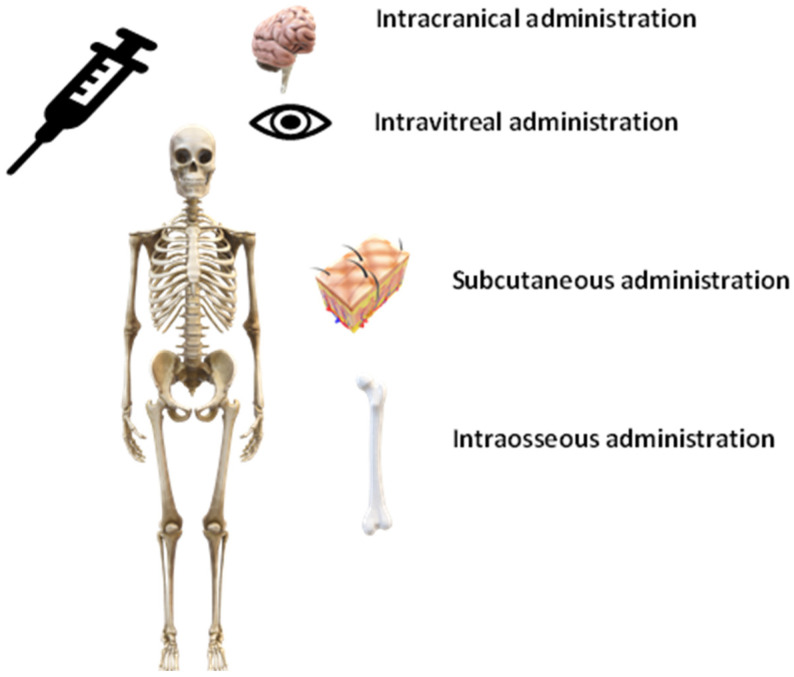
Overview of several possible methods of non-oral administration of encapsulated agents or cells within personalised therapies.

**Table 1 pharmaceuticals-16-00362-t001:** Recently published preparations of systems with encapsulated agents using different techniques.

Technique(s) Used	Encapsulated Agents	Coating Materials for Encapsulation	Application Areas	References
spray drying	resveratrol	polysaccharide—chitosan	intranasal drugs	[64]
	budesonide or rifampicin	oligosaccharide—lactose	intranasal drugs	[65]
	*Lactobacillus Acidophilus*	milk proteins, polysaccharides—pectin and maltodextrin	functional food	[66]
	*Lactobacillus plantarum*	polysaccharides (extract from Aloe vera)	functional food	[67]
	oleoresin from paprika	polysaccharides—gum arabic, starch	food supplement	[68]
(spray) freeze-drying	paclitaxel and doxorubicin	liposome	anticancer treatment	[69]
	ciprofloxacin	liposome	intranasal drugs	[70]
	turmeric oleoresin	protein-gelatine	food supplement	[71]
emulsion techniques	norcantharidin	liposome-emulsion hybrid delivery system	anticancer treatment	[72]
	Doxorubicin (Adriamycin)	polysaccharides—nanocellulose	anticancer treatment	[73]
	Zanamivir	polysaccharides—cellulose, gum arabic	intranasal drugs	[74]
	chlortetracycline	polysaccharides—starch, xanthan gum	model drug preparation	[75]
	magnesium	plant oil, lentil flour	functional food	[76]
	*Bifidobacterium lactis*	plant oil, beeswax	functional food	[77]

**Table 2 pharmaceuticals-16-00362-t002:** Overview of the published use of capsules for encapsulation.

Encapsulated Agent	Intended Use	Reference
Plant essential oils (rich in terpenes and terpenoids), extracted from thyme, oregano, lemongrass)	Antibacterial and antioxidant agents	[84]
Phenolic compounds extracted from onion	Antioxidant agent	[85]
Phenolic compounds extracted from bilberry	Antioxidant agent	[86]
Pea protein	Food (encapsulation due to taste masking)	[87]
*Lactobacillus fermentum* strain UCO-979C	Inhibition of *Helicobacter pylori*	[88]
*Lactobacillus plantarum* F1, *Lactobacillus reuteri* 182, *Lactobacillus helveticus* 305	Alginate capsules with reduced mortality of the cells during gelation	[89]
Vitamin D3	Food (enhanced D3 stability)	[90]
Essential oil encapsulated in yeast cells	Improved stability of essential oil	[91]

**Table 3 pharmaceuticals-16-00362-t003:** Some published uses of multiple emulsions for encapsulation.

Encapsulated Agent	Intended Use	Reference
Polyphenols	Prevention of ageing, cancer, inflammation and neurodegenerative diseases	[93,103,104]
Phenolic compounds	Antioxidants	[105]
Diclofenac sodium	Anti-inflammatory agent	[106,107]
Living cells	Cell therapy for regenerative, reproductive and transfusion medicine	[108]
Insulin	Diabetes treatment	[109]
Bioactive proteins	Functional food	[102]
Andrographolide (diterpenoid lactone)	Formulation with hepatoprotective activity	[110]
Bifonazole [1-[[1,1′-biphenyl)-4-phenylmethyl]-1H-imidazole)	Topical delivery of bifonazole to maximize its efficacy	[111]

**Table 5 pharmaceuticals-16-00362-t005:** Overview of the recent published uses of liposomes.

Encapsulated Agent	Intended Use	Reference
Catalase (EC 1.11.1.6)	Cancer therapy	[130]
Herbal phytochemicals (quercetin, vinblastine, hesperidin etc.)	Food supplements	[131]
Vincristine	Cancer therapy	[132]
Non-steroidal anti-inflammatory treatment	Drugs	[133]

## Data Availability

Not applicable for this work.

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
