# Peer review of "Encapsulation: A Strategy to Deliver Therapeutics and Bioactive Compounds?"

_pharmaceuticals, 2023, doi:10.3390/ph16030362_

Round 1

Reviewer 1 Report

Iveta Klojdová et al. provided a summary of recent developments in the use of encapsulation in the fields of medicine, dietary supplements, and functional foods, focusing on its advantages in targeted and supportive treatments. Even through the topic's interest, the following items should be adressed:

1. The text should generally be thoroughly proofread for spacing and a few minor grammatical errors.

2. Author should illustrate the encapsulation of drugs in a repressive table showing the name of the encapsulated drug and its application.

3. New title should be address "The recent applications of drug encapsulations"

4. New title should be address" Limitation of drug encapsulations"

Author Response

Thank you for your valuable comments and proposals! Please, see the attachment to view our responses to all queries. 

Reviewer 2 Report

The authors have reviewed the role of encapsulation towards the delivery of therapeutics and bioactive compounds. This is interesting work; however the following comments should be addressed by the authors before proceeding to the next step

1.    Rewrite the abstract and explain about the knowledge gap filled.

2.    In the introduction section the author should include the statement why encapsulation is required for delivering drugs.

3.    Page 2, 54 Clarify the statement, “the administration of encapsulated cells is a form of cell surface modification

4.    The quality of the Figure 1 & Figure 2 should be improved.

5.     The topic Cells given under subsection 2.1.2, should be provided separately after section. 2.2.

6.    Provide more in-depth discussion by citing previously published work related to the topic.

7.    Add one or two table to make a summary on different types of materials and the techniques used for encapsulation.

8.    The language of the manuscript has to be improved

9.    The plagiarism check using Turnitin showed a similarity index of 26 %

Author Response

(The authors gave the same response as above.)

Reviewer 3 Report

In this manuscript, the authors reviewed the different encapsulation techniques for the delivery of bioactive compounds. The manuscript was written well. Following are my queries and suggestion:

Ø  In Line 12-13, 42 and 132, authors stated encapsulation can modify the metabolism of bioactive compound. What is the role of encapsulation other than improving dissolution behaviour (or protecting from gastric environment) in the GI tract.

Ø  In Fig 2: arrows for capsules were pointed from high pressure homogenization and sonication. Authors need to correct this.

Ø  Section 3.3: Authors can also review the spray freeze drying technique, nanospray dryer and electrospraying technique. These novel techniques are recently used for producing encapsulated particles. 

Ø  Authors can add a section for “fortification of these encapsulated particles into food matrix”. 

Ø  The title of Section 4 is “Behavior and metabolism of dispersions in human body”. However, the concept of metabolism is not discussed anywhere. 

Author Response

(The authors gave the same response as above.)

Round 2

Reviewer 1 Report

The authors have addressed all my concerns, and I recommend acceptance of the paper.

Author Response

We would like to thank you for your valuable remarks and proposals that significantly improved the quality of our manuscript, and for your time devoted to reviewing it. 

Reviewer 2 Report

The authors have addressed most of my concerns, however, I have suggest the authors to cite the following articles

https://doi.org/10.3390/polym14050950 

https://doi.org/10.3389/fonc.2022.891673

https://doi.org/10.1016/j.mtcomm.2022.105124

Author Response

(The authors gave the same response as above.)

Reviewer 3 Report

Authors modified the manuscript and responded the queries. 

Author Response

(The authors gave the same response as above.)
